# Experimental Investigation and Fault Diagnosis for Buckled Wet Clutch Based on Multi-Speed Hilbert Spectrum Entropy

**DOI:** 10.3390/e23121704

**Published:** 2021-12-20

**Authors:** Jiaqi Xue, Biao Ma, Man Chen, Qianqian Zhang, Liangjie Zheng

**Affiliations:** 1School of Mechanical Engineering, Beijing Institute of Technology, Beijing 100811, China; 3120185226@bit.edu.cn (J.X.); mabiao@bit.edu.cn (B.M.); qianqian.zhang1995@gmail.com (Q.Z.); 3120170250@bit.edu.cn (L.Z.); 2Key Laboratory of Science and Technology for National Defense, Beijing Institute of Technology, Beijing 100811, China; 3Collaborative Innovation Centre of Electric Vehicles in Beijing, Beijing 100081, China

**Keywords:** wet clutch, buckling, entropy, fault diagnosis

## Abstract

The multi-disc wet clutch is widely used in transmission systems as it transfers the torque and power between the gearbox and the driving engine. During service, the buckling of the friction components in the wet clutch is inevitable, which can shorten the lifetime of the wet clutch and decrease the vehicle performance. Therefore, fault diagnosis and online monitoring are required to identify the buckling state of the friction components. However, unlike in other rotating machinery, the time-domain features of the vibration signal lack efficiency in fault diagnosis for the wet clutch. This paper aims to present a new fault diagnosis method based on multi-speed Hilbert spectrum entropy to classify the buckling state of the wet clutch. Firstly, the wet clutch is classified depending on the buckling degree of the disks, and then a bench test is conducted to obtain vibration signals of each class at varying speeds. By comparing the accuracy of different classifiers with and without entropy, Hilbert spectrum entropy shows higher efficiency than time-domain features for the wet clutch diagnosis. Thus, the classification results based on multi-speed entropy achieve even better accuracy.

## 1. Introduction

The wet clutch is widely used in automatic transmission systems and plays a vital role in torque transfer [1]. As shown in Figure 1, a multi-disc wet clutch consists of friction disks and separate disks. The friction disks are splined to the driving shaft by their inner splines, while the separate disks are mounted to the cylinder liner. During the engagement process, the separate disks (driven parts) and friction disks (driving parts) are pressed together by the piston to synchronize the speed. Usually, the engagement process completes within a second, so the friction between disks could generate tremendous heat in the system very quickly, which could potentially cause the deformation of the disks [2]. The deformation or buckling of the friction components is illustrated in Figure 2; the inner ring of the buckled friction disk has an apparent gap from the ground, and the surface is curved. 

Many researchers have studied the buckling problem in the wet clutch system. Zagrodzki [3] solved temperature fields of clutch disks during wet clutch engagement using numerical methods in the 1980s, and he pointed out a particular failure pattern of the disks related to temperature field, namely dishing. This typical buckling is shown in Figure 2. Zagrodzki and Truncone [4] presented the formation of hot spots on the disks due to uneven contact. Li et al. [5] investigated buckling-induced thermoelastic instability (TEI) via a bench test. They considered the impact of the spline teeth and built up a buckling model. Zhao et al. [6] proposed a contact model to demonstrate the impact of the buckling on the temperature field, and their study reveals the huge temperature variance between contact and non-contact areas. Yu et al. [7] discussed the potential damage caused by the disk buckling, and they concluded that the deformation of disks would create not only huge temperature variance but also considerable friction torque between contact and non-contact areas, which will escalate the deformation in return. From the above studies, it can be concluded that the buckling of the disks can affect the performance and impair the life of the wet clutch, and the damage increases along with the buckling degree. Therefore, it is reasonable to quantify the buckling level as a health indicator for the wet clutch system.

From the maintenance point of view, detecting the disk deformation in time is necessary. Prognostics health management (PHM) for engineering equipment has been developing quickly in recent years, and many researchers have utilized artificial intelligence (AI) techniques for fault diagnosis [8]. Anyi et al. [9] built up a PHM model based on Cuckoo Search and a back-propagation network for a transformer system; the method is efficient for real operation conditions. Barbieri et al. [10] analyzed different signals using techniques based on wavelet transform, mathematical morphology, and energy entropy to verify the presence of damage in the automobile gearbox. Nguyen et al. [11] constructed a speed invariant, stacked, sparse, autoencoder deep neural network for the gearbox with improved computational capabilities, and the results outperformed the conventional methods. Liu et al. [12] proposed an efficient approach to identify the real-time problem based on multi-scale permutation entropy. Sun et al. [13] developed a deep neural network for classification of induction motor faults, in which a sparse autoencoder is applied to improve the robustness of feature representation. A novel fault diagnosis method for a hydraulic pump based on a symbolic perceptually important point is discussed in [14], and the method shows excellent results for machines in harsh conditions. Lu et al. [15] investigated the effectiveness of the stacked denoising autoencoder (SDA) for health state identification of rotary machinery components. Antonio et al. [16] improved permutation entropy interpretation by identifying artefact information and providing a proper linear filter framework. With the combination of wavelet transform, neural networks, and machine learning techniques, Chen et al. [17] presented a comprehensive method and successfully classified faults with very similar signal features via CNN and ELM. Yang et al. [18] used a mutual information-sample entropy-based scheme to extract the weak fault of hoist bearing in a strong noise environment. By extracting the feature vectors of thermal images taken from devices, Glowacz [19] developed a diagnosis method using imaging processing and identified faults effectively.

Although the fault diagnosis methods for rotary machineries such as bearing, gearbox, and wind turbines have made progress, relevant applications for wet clutch systems have barely been studied. One obstacle to the fault diagnosis for the wet clutch system lies in feature selection. Fault diagnosis methods for rotary machinery have some clear patterns, for example, the bearing system can be classified based on bearing fault frequencies [20]. However, the operation states of the wet clutch are complex to formulate, and the instability of the multi-disc wet clutch under high speed has not yet been clarified [21]. The coefficient of friction (COF) has been adopted for clutch monitoring, but it is impossible to implement in real-life applications [22]. In this paper, the authors adopt the rub impact phenomenon of the disks in the wet clutch during the detached state as the intuition for fault diagnosis. This phenomenon was studied by Hou et al. [23] via an open wet clutch test rig; their work revealed the phenomenon of disk wobbling and confirmed the existence of mechanical impact between the disks. Moreover, when the rotation speed increases, the wobbling and impact became non-linear. Hu et al. [24] studied the rub impact in their research about drag torque, and they found a strong correlation between the drag torque rise and the rub impact effect of the disks at a high rotation speed.

This paper presents a fault diagnosis method for a multi-disc wet clutch based on time-frequency entropy at varying speed. Some major contributions can be summarized as follows: (1) The multi-disc wet clutch is crucial to transmission systems, but almost zero attention has been paid to fault diagnosis methods for the wet clutch. The proposed paper is a pioneer in developing fault diagnosis models and successfully classifying the wet clutch fault. (2) As for feature selection, traditional time-domain features fail to classify the fault of the wet clutch effectively due to the complexity of the clutch structure and operation mechanism; this paper adopts Hilbert spectrum entropy to identify the tiny differences caused by faults in the system. (3) Most fault diagnosis methods utilize the signal features at the same speed. In the proposed methods, the dimension of the data is the signal features at varying speeds rather than different features at the same speed, which can take advantage of the fault-induced vibration varying along with rotation speed. 

The rest of this paper is organized as follows. Section 2 starts with a brief introduction to rub impact among clutch disks and how buckling can affect this phenomenon, and then the experimental objects are classified into four groups based on the health states, and a bench test is conducted to collect the vibration signal for all groups at varying operation speed. Section 3 introduces how raw data are processed via Hilbert–Huang transform to generate the Hilbert spectrum and corresponding time-frequency entropy. Section 4 presents a fault diagnosis method for the multi-disc wet clutch system based on Naïve Bayes and compares time-frequency entropy’s effectiveness with other time-domain features as the input quantities. Conclusions and final remarks are given in Section 5.

## 2. Preliminaries and Bench Test

### 2.1. Wobbling and Rub Impact of the Disks in the Wet Clutch

This paper mainly adopts the wobble and rub impact of the clutch disks as the initial criteria for distinguishing the signal with different buckling degrees. As shown in Figure 3, during the disengagement state, instead of rotating at the same position on the *Z*-axis, the separate disk and friction disk usually have other motions, such as translating along the *Z*-axis, and oscillating around the *X*-axis and *Y*-axis [24].

The wobble effect is explained in Figure 4, assuming the separate disk oscillates and the adjacent friction disks remain vertical. When the angle reaches a certain amount, the edge of the disk can impact another, and the disk could bounce back to hit another side, so the separate disk is wobbling between friction disks. Hou [23] found out in an open bench test that the intensity of the impact depends on the gap between the disks; the smaller the gap is, the smaller the impact angle for collision, and the more evident the impact phenomenon becomes. As for the buckled disk, compared with the healthy flat disk, the buckling shortens the effective clearance between the disks and reduces the impact angle, so more evident impact could happen to the buckled disks. Therefore, the wet clutch with higher buckling degree is likely to generate more chaotic vibration signals.

### 2.2. Health State Classification

Before the bench test, four pairs of separate disks and friction disks are selected depending on their buckling degree. Then every single disk is set on a smooth surface (the outer ring is contacting the surface) and scanned via HandyScan3D. After importing the data into the computer, their surface topography can be read in Geomagic. The steps can be seen in Figure 5. The scan result of one buckled separate disk is shown in Figure 5c; the measured surface height increases from the outer to the inner ring while it merely changes circumferentially. Because the buckling can lead to an increase in the height of the inner ring, for each disk, the absolute heights of six red dots are measured and compared with the thickness of the healthy flat disks. The more the disks buckled, the larger the value that will be measured. The reference thickness of the separate disk is 2 mm and of the Cu-based friction disk is 3 mm.

Four groups have been marked as A, B, C, and D based on their measured and calculated average heights of the six points. The parameters are listed in Table 1. Class A represents the healthy condition, where separate disk and friction disk are close to original values. Classes B and C are in slight and medium buckling states, while Class D is severe buckling. For comparison, the total occupation of Class D is 15 mm, nearly triple the space of Class A.

### 2.3. Bench Test

The arrangement of the test rig is shown in Figure 6. The rig involves a transmission line and a signal collection system. The transmission system consists of an electromotor, a speedometer, a clutch pack, a torque meter, flying wheels, and a braking system. During the operation, the separate disks are held still to the right side of the clutch pack by the braking platform, while the friction disks are driven and rotating as they are mounted on the driving shaft by the inner spline. The motor is controlled by a PC. When the motor starts, the rotation of the motor could simulate the speed variance between the friction disks and the separate disks during the disengagement stage. In the experiment, because the disks near the piston are more likely to buckle than the distant disks [25], the position of tested objects is set next to the circlip as circled in red in Figure 6, and the rest of the disks are in healthy state. One thing to mention is that the disk pairs are taken from real vehicles. The material of the separate disk is 65 Mn, and the friction disk is an iron core disk coated with Cu.

The signal collection system used in the experiment is provided by the B&K company, including a transducer, a signal pre-processing acceptor, and a PC. The transducer is a B&K 4518-03 accelerometer mounted axially to the transmission system outside the clutch box. The data are collected at a rate of 64 kHz and processed via B&K PULSE software. During the experiment, the lubrication oil (Automatic Transmission Fluid, ATF) is set to be at ambient temperature. Data are recorded with an increment of 100 rpm ranging from 500 rpm to 1100 rpm (7 different rotation speeds), and each rotating speed lasts 5 s. The experimental parameters are listed in Table 2.

## 3. Signal Processing: Hilbert Spectrum and Time-Frequency Entropy

### 3.1. Hilbert–Huang Transform and Hilbert Spectrum

The vibration signal of each fault type at 700 rpm is plotted simply in Figure 7. As can be seen, it is impossible to figure out specific patterns and differences between each type from the time domain. Since the signal of the wet clutch is more chaotic than other rotating machinery [23], the Hilbert–Huang transform is introduced here for signal processing because it has excellent characteristics to analyze non-stationary and non-linear signals [26]. This technique was improved by Huang, who introduced empirical mode decomposition (EMD) before the signal undergoes Hilbert transform [27]. The flow chart of the Hilbert–Huang transform is shown in Figure 8.

Before applying the Hilbert transform, the whole signal needs to be decomposed into several IMFs (intrinsic mode oscillations); this is the empirical mode decomposition (EMD). The IMFs need to meet several requirements: they are defined as functions having the same (or differing at most by one) numbers of zero-crossing and extrema, and also having symmetric envelopes (with respect to the time axis) defined by the local maxima and minima, respectively. After EMD, a signal *x*(*t*) can be expressed as:(1)$$x\left(t\right)=\overset{n}{\underset{j=1}{\sum }}I_{j}+r_{n}$$
in which $I_{j}$ is the jth decomposed IMF of the original signal $x\left(t\right)$, $r_{n}$ is the residual signal which can be a constant or the mean trend of the signal. Then, the Hilbert spectrum can be expressed:(2)Hω,t=Re∑i=1naitej∫ωitdt

Re is the operator of the real part, and $a_{i}\left(t\right)$ and $\omega_{i}$ represent the amplitude and instantaneous frequency, respectively. Note that the residual term $r_{n}$ is neglected as it only represents very little energy. Taking the signal of Class A at the speed of 700 rpm as an example, the process of generating the Hilbert spectrum is shown in Figure 9. By doing so, the time domain signals are transferred to spectra that contain energy information.

Figure 10 shows the Hilbert spectrum of four classes at 700 rpm. In the figures, all the spectra are at the same energy scale, the *X*-axis represents the time, and the *Y*-axis represents the frequency. The local color represents the power intensity of the vibration at a specific time and frequency. From the figure, the energy density of the four appears different. For example, the intensity of the bright dot in Classes B, C, and D is greater than that of Class A, which indicates that at 700 rpm, the signal is more chaotic as the buckling degree increases. One thing that should be mentioned is that because the wet clutch is a complex assembly with many other parts and ATF cooling inside, the vibration variance of the four classes may not be entirely related to the rub impact of the clutch disks, but it is clear that the buckling variance has increased the vibration variance on the spectrum plots. To continue, although the spectra of different categories show a little difference, each spectrum needs to be assigned an identical value for further classification.

### 3.2. Time-Frequency Entropy

In order to quantify the signal variance caused by the buckling of friction components, spectrum entropy is introduced and calculated here. The block diagram is presented in Figure 11 to illustrate the process. After Hilbert transform, the spectra are transformed into gray-scale images for calculation convenience. In this paper, the authors have filtered out the signal below 1 kHz (the main noise with large power in the gearbox is below 1 kHz [28]) and above 3 kHz (“blank space” with no power starts to dominate over 3 kHz). So, the spectrum entropy of 1 kHz to 3 kHz is used in this paper.

As illustrated in Figure 11, once the gray-scale image is obtained, the entropy can be determined. Firstly, the spectrum is equally divided into N small blocks, and the energy of each block can be calculated depending on its gray value (the value ranges 0–255 from black to white) and marked as $W_{i}\left(i=1,2\ldots N\right)$. Then, the total energy of the spectrum plane can be calculated as *A*. The energy for all blocks is normalized, and the normalized value for each block is written as:(3)$$q_{i}=\frac{W_{i}}{A}$$

Finally, the total time-frequency entropy of the vibration signal can be expressed as:(4)$$s\left(q\right)=-\overset{N}{\underset{i=1}{\sum }}q_{i}ln\;q_{i}$$
where $\sum_{i=1}^{N}q_{i}=1$, which satisfies the original normalization criterion of information entropy calculation; N=256 in the present work, which means for each signal piece, the Hilbert spectrum is equally divided into 256 blocks for entropy estimation. The larger the value of $s\left(q\right)$, the more the energy due to vibration and the greater the probability of rub impact. 

The present work has calculated the Hilbert spectrum entropy of each group from 500 rpm to 1100 rpm, and plotted the fitted curves as shown in Figure 12. From the figure, the entropy shows some vibration characteristics of the system. Class B has larger response entropy amplitudes at 700 rpm, while Class C takes over at 800 rpm. Another thing that can be utilized for further machine classification is that despite some values being very close, the class with higher buckling degree generally shows higher Hilbert spectrum entropy. For instance, the entropy in Class D is larger than Class A at almost any rotation speed.

## 4. Entropy-Based Fault Diagnosis Method

The framework of the fault diagnosis method is depicted in Figure 13. The first two parts of the framework have been demonstrated in Section 2 and Section 3, respectively. Since the key point of this work is to develop a fault diagnosis system for the wet clutch based on time-frequency entropy rather than developing machine learning or feature selection techniques, in this paper, Naïve Bayes, KNN, and SVM algorithms are simply chosen as the classification methods. Meanwhile, some commonly used time-domain features are selected as model inputs. Firstly, the inputs are the features at the same speed, and the accuracies with and without entropy are compared. Then, diagnosis models with entropy at different rotating speeds are proposed, and the results are discussed at the end of this section.

### 4.1. Classifiers

#### 4.1.1. Naïve Bayes Classifier

The Naïve Bayes method is a well-known classifier based on Bayes’ theorem and conditional independence assumption. For a given data set $T=\left\{\left(x_{1},y_{1}\right),\left(x_{2},y_{2}\right),\ldots ,\left(x_{n},y_{n}\right)\right\}$, $X=\left\{x_{1},x_{2},\ldots ,x_{n}\right\}$ defines the attributes and $Y=\left\{y_{1},y_{2},\ldots ,y_{n}\right\}$ defines the label. The general form of conditional independence assumption if X contains n attributes is written below:(5)$$P\left(X_{1},X_{2},\ldots X_{n}|Y\right)=\overset{n}{\underset{i=1}{\prod }}P\left(X_{i}|Y\right)$$

Notice that when Y and $X_{i}$ are Boolean variables, we need only 2 n parameters to define $P\left(X_{i}=X_{ik}|Y=X_{j}\right)$ or the necessary *i*,*j*,*k*. This is a dramatic reduction compared with the $2\left(2^{n}-1\right)$ parameters needed to characterize $P\left(X|Y\right)$ if we make no conditional independence assumption. Assuming *Y* is any discrete value and the attributes $X_{1},X_{2},\ldots X_{n}$ are any discrete or real values, the goal of the classifier is to output the probability distribution *Y* for each given instance *X*. Assuming that $X_{I}$ are conditionally independent given Y, the expression for the probability that *Y* will take on its kth possible value, according to Bayes’ rule, is:(6)$$P\left(Y=Y_{k}|X_{1},X_{2},\ldots X_{n}\right)=\frac{P\left(Y=Y_{k}\right)\prod_{i=1}^{}P(X_{i}|Y=Y_{k})}{\sum_{j}^{}P\left(Y=Y_{j}\right)\prod_{i=1}^{}P(X_{i}|Y=Y_{k})}$$
(7)$$Y\leftarrow arg\underset{Y_{k}}{\max}\;P\left(Y=Y_{k}\right)\overset{}{\underset{i=1}{\prod }}P\left(X_{i}|Y=Y_{k}\right)$$

#### 4.1.2. k-Nearest Neighbor

KNN is an instance-based learning algorithm, which searches for the most similar eigenvectors of the instance in the database [29]. For a training sample $\left(x,y\right)$, the KNN algorithm searches for the *k* nearest instances to *x* based on a setting distance. The neighborhood containing these *k* instances is represented by $N_{k}\left(x\right)$. Then, the label of test sample *x* can be calculated based on decision rules:(8)$$Y=arg\underset{Cj}{\max}\;\underset{xi=N_{k}\left(x\right)}{\sum }I\left(y_{j}=c_{j}\right)\;i=1,2,\ldots ,N;j=1,2,\ldots ,K$$
where $x_{i}$ is the feature vector of the unlabeled instance, I is the indicator function, and $y_{j}=c_{1},c_{1},\ldots ,c_{K}$, is the label. Compared with other algorithms, KNN is simple to apply.

#### 4.1.3. Support Vector Machine

SVM is a computational learning method for classification of small samples [30]. SVM works very well if feature vectors are linearly separable. Algorithmically, it separates the data samples through the hyperplane with the greatest margin, and the hyperplane $f\left(x\right)$ is: (9)$$y=f\left(x\right)=W^{T}\;x+b=\overset{N}{\underset{i=1}{\sum }}W_{i}x_{i}+b$$
where W is an N-dimensional vector, and b is a scalar. SVM does not work well for many overlapping classes and signals with high noise.

### 4.2. Features in the Time Domain

Many researchers have taken the time-domain features as the input for fault diagnosis methods of rotation machinery [31,32]. Similarly, the present work also selects some time-domain features for fault diagnosis, namely, mean root square, peak value, crest factor, kurtosis, and skewness, and they are listed in Table 3.

In the table, the root mean square ($X_{rms}$) value reflects the mean energy of the signal. The peak value is the largest measured amplitude in a signal. The crest factor or peak-to-average ratio (PAR) is calculated from the peak amplitude of the waveform divided by the RMS value. Kurtosis is the statistical moment of the probability density function. The skewness value represents the measure of the degree of asymmetry of distribution; it may be positive, or negative, or zero.

### 4.3. Results and Discussion

Data are labeled as A, B, C, and D for different buckling degrees. Firstly, all signal clips are split into 40 smaller pieces (7 different speeds, so each group has 40×7=280 pieces; each piece has around 8000 sampling points) for machine learning usage. Then, the feature values (time-domain and entropy) of the signal pieces are calculated. Firstly, the inputs are the features at the same speed, and the accuracies with and without entropy are compared. Then, the diagnosis models with entropy at different rotating speeds are presented and analyzed. A *k*-folds approach for training and testing is used with *k* = 5.

After the testing process, validation of the classification is done, and the efficiencies of the classifiers for different features are obtained and compared. Table 4 provides the overall accuracy of three different classifiers with the same rotation-speed input features, and the better classification accuracy is marked in a bold font. Without a doubt, the model with entropy input overwhelmed the models that only have time-domain inputs. It also can be seen that none of the accuracies with time-domain features is greater than 70%, which suggests that time-domain features have limited success in identifying the wet clutch system. In particular, the entropy improves the accuracy at 500 rpm and 700 rpm drastically; it improves the testing accuracy by nearly 20% and 10% compared with the system without entropy, respectively. Additionally, for any classifier, the accuracy at 600 rpm is relatively low, while 1100 rpm has overall better performance. 

Since the Hilbert spectrum entropy can better reflect the system characteristics, it positively influences the classification accuracy. This paper tests the methods using the entropy at different speeds as the inputs, and the test results are shown in Table 5. All three classifiers achieve nearly 90% accuracy, outperforming any model using time-domain features. SVM has the best performance of 92.5%. 

The confusion matrices for the multi-speed entropy methods of the three classifiers are shown in Figure 14. Based on time-frequency entropy, all three methods are able to distinguish Class A with a true positive rate of 100%, bearing in mind that Class A is the only healthy group, which means when the inputs are multi-speed entropy, all methods can effectively distinguish the healthy state and will not misdiagnose the healthy state into any other buckling states. On the contrary, Class D has the lowest diagnostic accuracy in all three systems (75% for Naïve Bayes, 80% for KNN, and 82.5% for SVM); 12.5% of Class D are considered group C and 5% group B for both Naïve Bayes and KNN. This may be due to the similarity of the time-frequency entropy of buckling disks under some rotating speeds. The results of Classes B and C are relatively ideal. The mistakes made by different classifiers are quite similar. SVM outperforms the other two methods because it makes zero mistakes to classify Classes B, C, and D into Class A. From Class A to Class D, the prediction accuracy decreases as the buckling degree increases. Class C has more false predictions than Class B since the model misclassified more Class C into Class D. In addition, SVM has better performance than KNN and Naïve Bayes; it misclassified less Class D into other groups. On the whole, the decline in diagnostic accuracy indicates that a signal with a greater buckling degree is more challenging to identify. From the perspective of engineering applications, it is worthwhile to try to overcome this because people are looking for a method that alerts about faults. The more serious the fault, the more accurate the prediction results required. This limitation will be a focus of our future work.

## 5. Conclusions

This paper provides a fault diagnosis method for the wet clutch buckling problem based on Hilbert spectrum entropy. The rub impact and wobble effect of the clutch disk are introduced as an intuition to demonstrate the potential difference caused by buckling. Then, a multi-speed bench test is conducted, and the tested objects are divided into four classes based on buckling degree. Then, the data are processed via Hilbert–Huang transform to obtain the Hilbert spectrum. After converting the spectrum into gray-scale images, the entropy of the spectrum is calculated and stored as the input features. Finally, Naïve Bayes, KNN, and SVM are used to verify the efficiency of the entropy-based method compared with other time-domain features.

Hilbert spectrum entropy has a positive impact on the accuracy of fault diagnosis methods. From the results, compared with other time-domain features, the proposed entropy-based method drastically increases the accuracy. Meanwhile, entropy-based SVM has the best effectiveness to classify the clutch with different buckling degrees, especially in identifying Class A among other fault classes.

Also, the computation of the entropy suggests that the buckling of the disk could lead to an increase in vibration energy in the clutch pack. This provides valuable information for trend analysis in clutch buckling monitoring. One limitation of the classification model is that the accuracy in identifying a higher buckling degree of the system needs to be improved. Therefore, recognizing the fault frequency more effectively will be a focus of our future work.

## Figures and Tables

**Figure 1 entropy-23-01704-f001:**
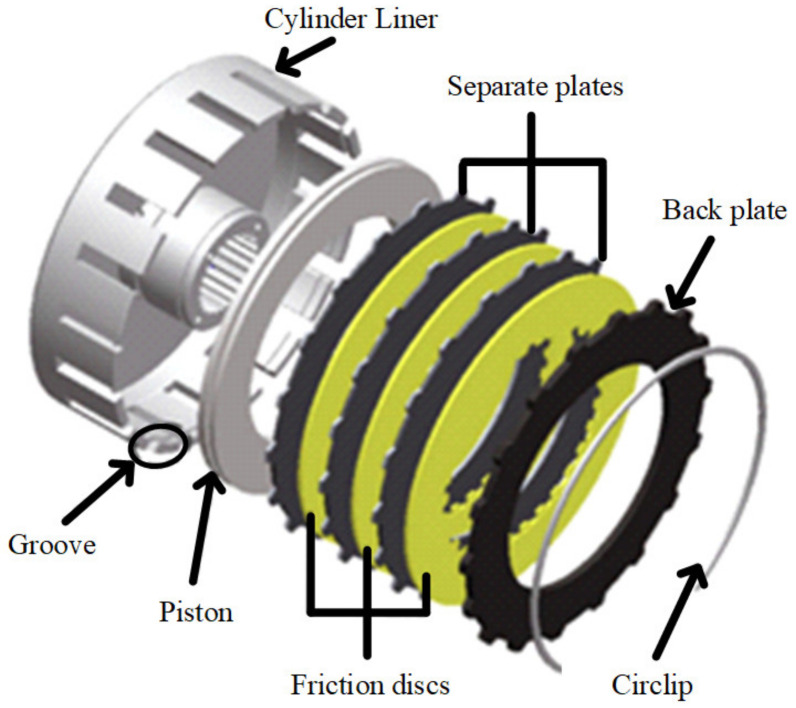
3D sketch of the clutch system.

**Figure 2 entropy-23-01704-f002:**
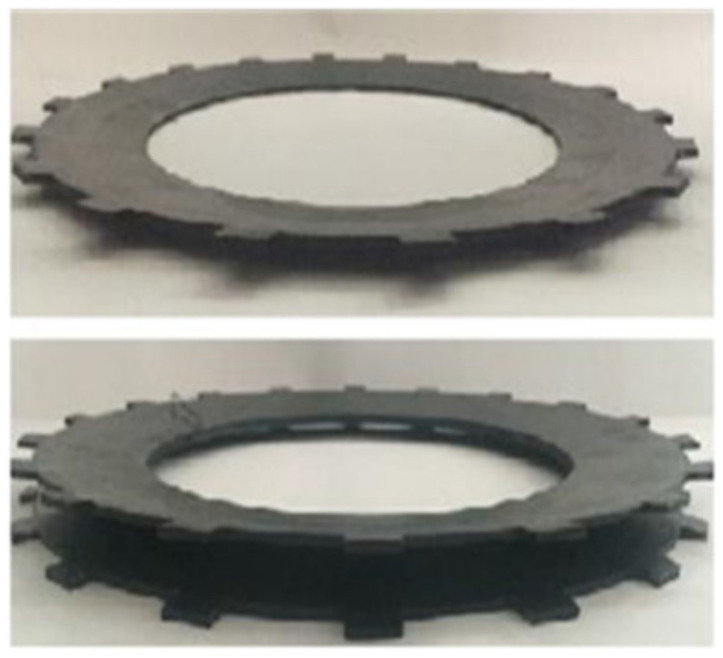
A typical disk buckling (dishing).

**Figure 3 entropy-23-01704-f003:**
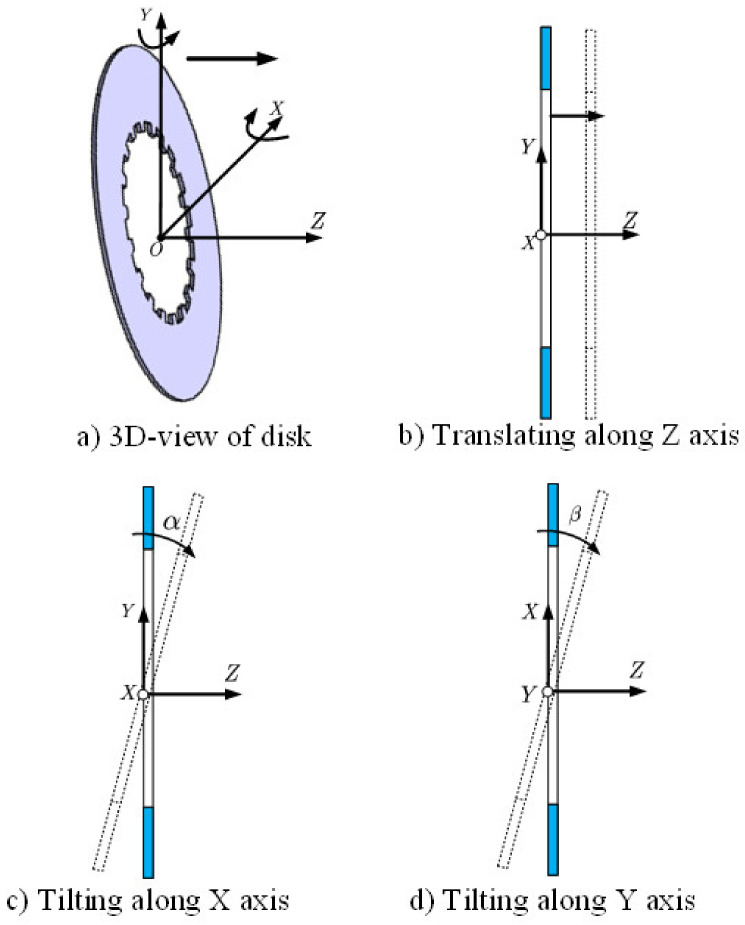
Potential motions of separate/friction disk.

**Figure 4 entropy-23-01704-f004:**
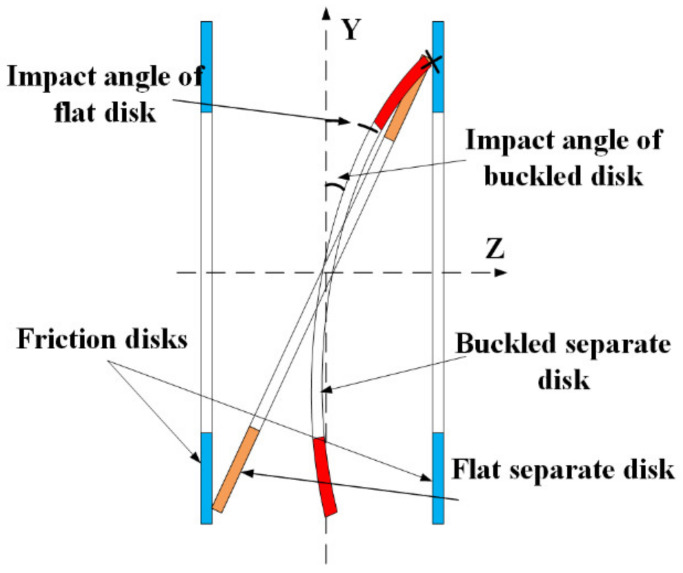
Wobbling effects of flat/buckled disks.

**Figure 5 entropy-23-01704-f005:**
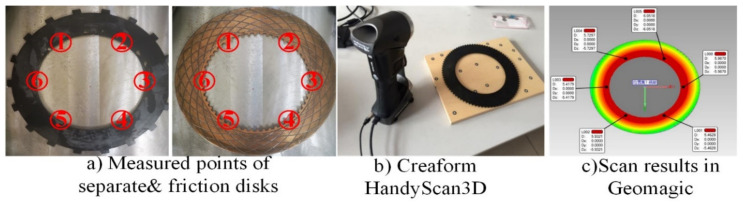
(**a**–**c**) Measured steps of the disk.

**Figure 6 entropy-23-01704-f006:**
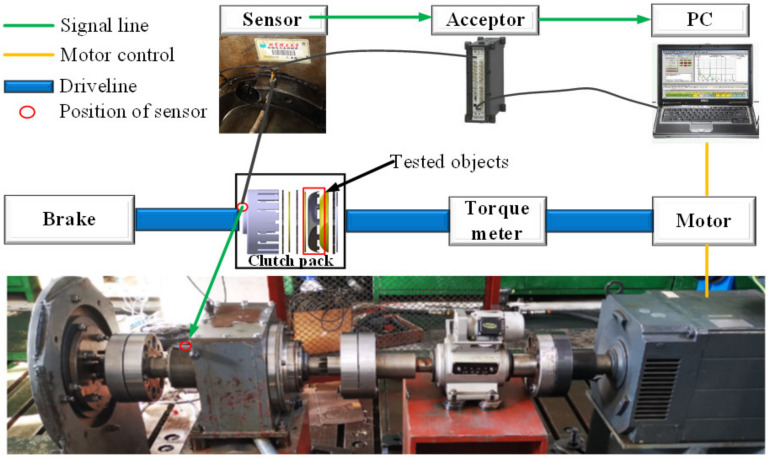
Test bench and signal acquisition system.

**Figure 7 entropy-23-01704-f007:**
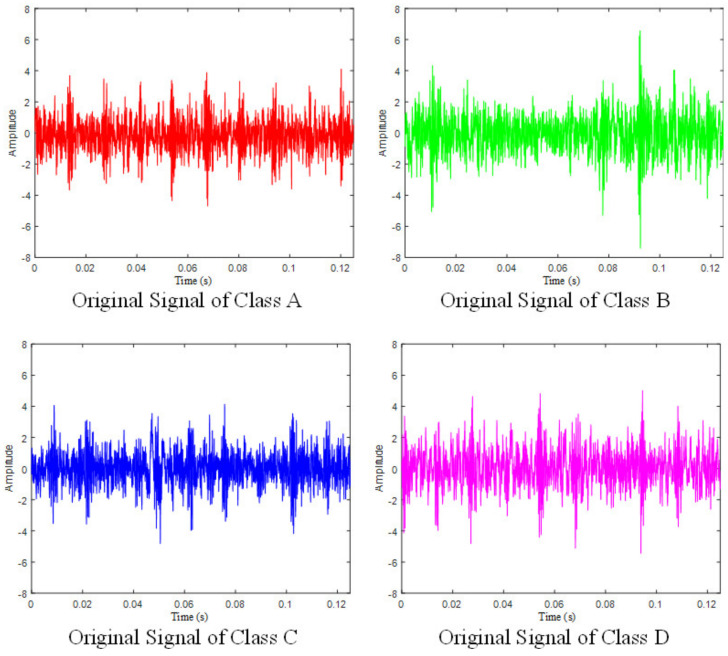
Original signal of four classes.

**Figure 8 entropy-23-01704-f008:**
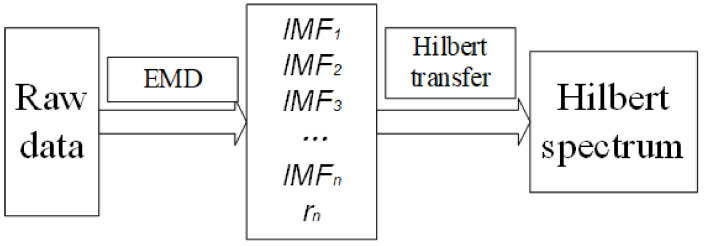
Flow chart of Hilbert–Huang transform.

**Figure 9 entropy-23-01704-f009:**
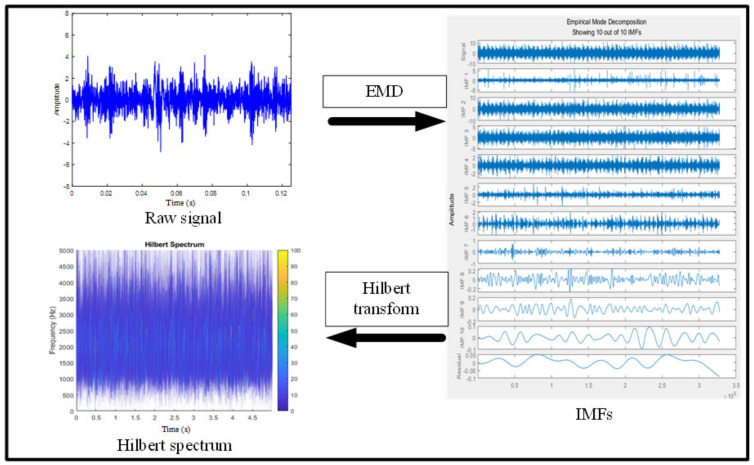
Hilbert–Huang transform flow chart for Class A at 700 rpm.

**Figure 10 entropy-23-01704-f010:**
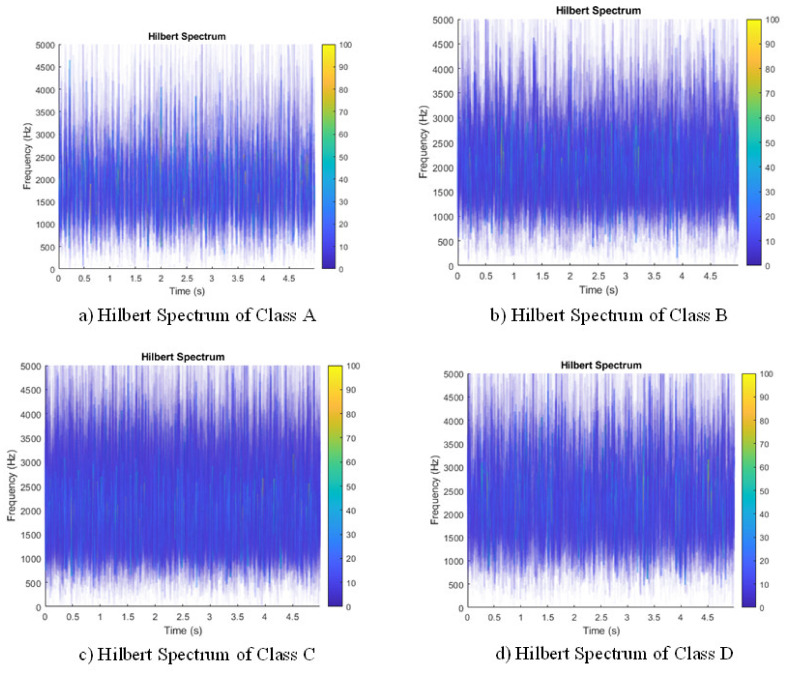
(**a**–**d**) Hilbert spectra of four groups at 700 rpm.

**Figure 11 entropy-23-01704-f011:**
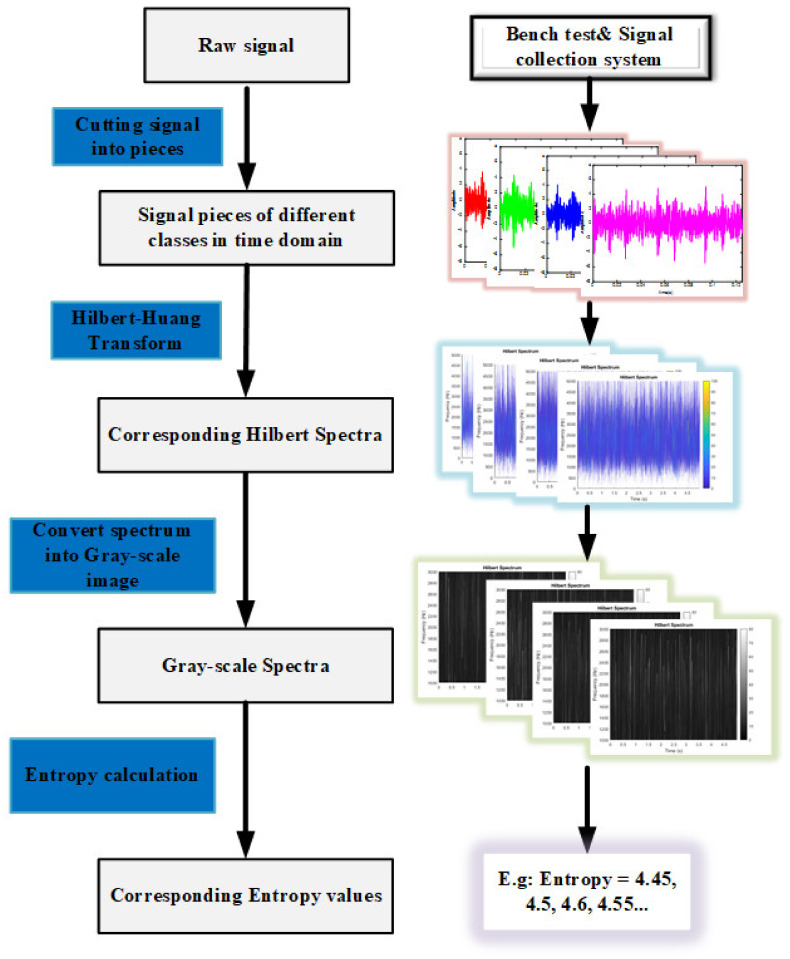
Determination of the spectrum entropy.

**Figure 12 entropy-23-01704-f012:**
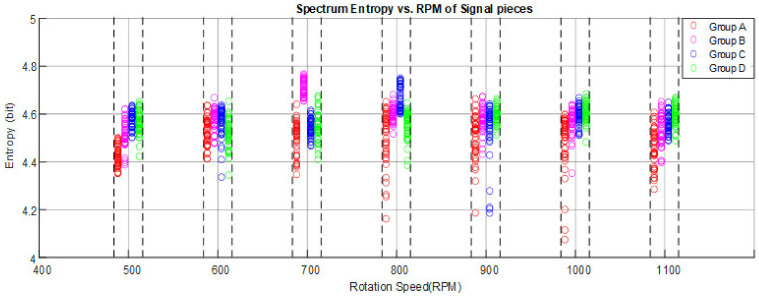
The entropy distribution vs. RPM of the four classes.

**Figure 13 entropy-23-01704-f013:**
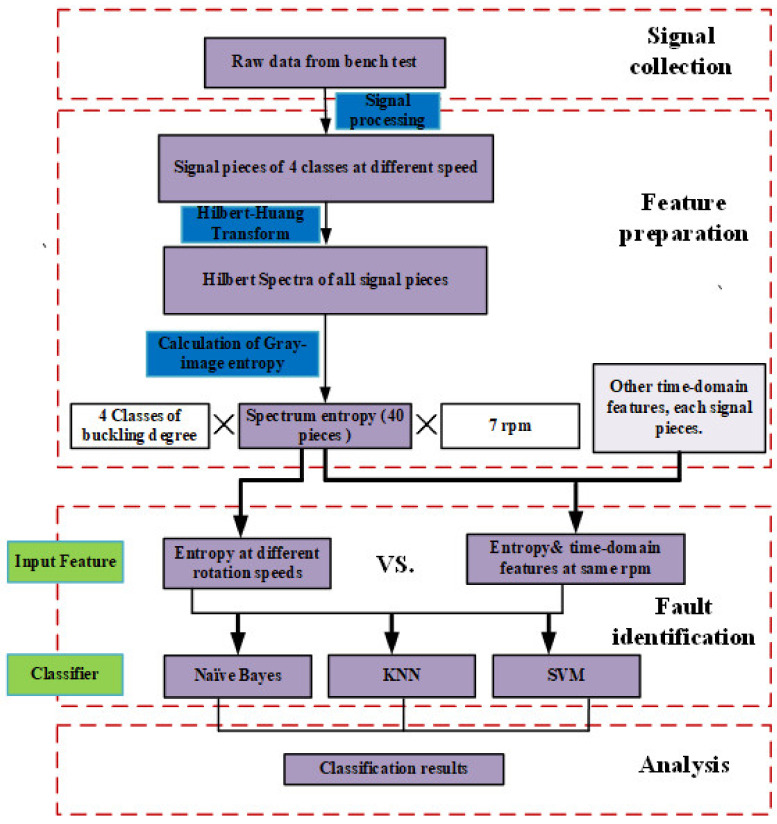
Framework of the entropy-based fault diagnosis method.

**Figure 14 entropy-23-01704-f014:**
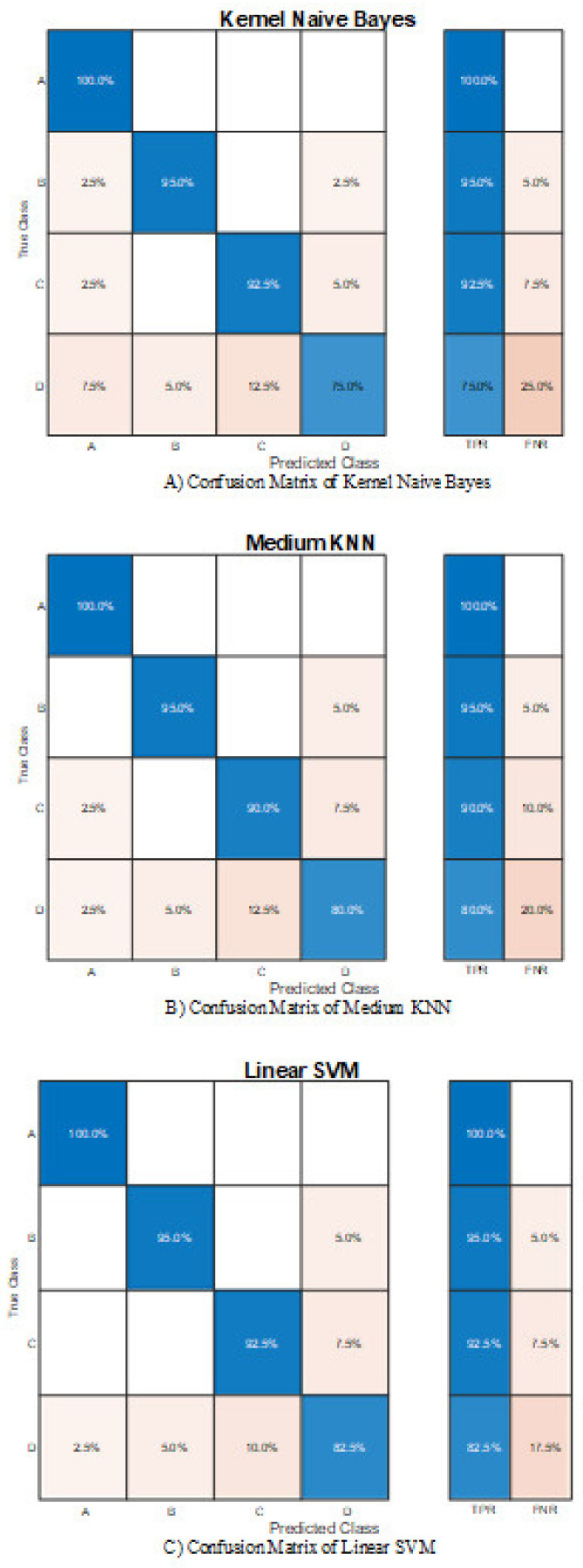
(**A**–**C**) Confusion matrices of entropy-based Naïve Bayes.

**Table 1 entropy-23-01704-t001:** Buckling state of four conditions.

Health State	Separate Disk (mm)			Friction Disk (mm)		
Class A Normal condition	2.03	2.15	2.08	3.05	3.22	3.09
	1.93	2.14	2.04	3.08	3.14	3.14
	Average: 2.06			Average: 3.12		
Class B Slight buckling	3.31	3.39	3.41	4.03	3.98	4.05
	3.33	3.48	3.36	3.95	4.01	3.97
	Average: 3.37			Average: 4.02		
Class C Medium buckling	4.17	4.21	4.20	4.57	5	4.75
	4.2	4.1	4.15	4.7	4.5	4.63
	Average: 4.17			Average: 4.69		
Class D Severe buckling	6.78	6.9	6.76	7.45	7.55	7.46
	6.79	6.78	6.86	7.26	7.3	7.32
	Average: 6.81			Average: 7.39		

**Table 2 entropy-23-01704-t002:** Experimental parameters.

Friction Component	Item	Outer Radius	Inner Radius	Healthy Thickness	Density
	Separate Disk	0.129 m	0.081 m	0.002 m	7800 kg/m^3^
	Friction Disk	0.125 m	0.086 m	0.003 m	5500 kg/m^3^
Operation parameter	ATF temperature	Rotation speed (rpm)			Sampling frequency/time
	30 °C	500 600 700 800 900 1000 1100			64 kHz/ 5 s per run

**Table 3 entropy-23-01704-t003:** Time-domain features used in machine learning.

Time Domain Feature	Expression
**Root mean square**	$X_{rms}=\sqrt{\frac{1}{N}\overset{N}{\underset{i=1}{\sum }}{\left(x_{i}-\bar{x}\right)}^{2}}$
**The peak value**	$X_{peak}=\max\left(x_{i}\right)$
**The crest factor**	$X_{p}=\frac{X_{peak}}{X_{rms}}$
**Kurtosis**	$X_{k}=\frac{\frac{1}{N}\sum_{i=1}^{N}{\left(x_{i}-\bar{x}\right)}^{4}}{X_{rms}^{4}}$
**Skewness**	$X_{s}=\frac{\frac{1}{N}\sum_{i=1}^{N}{\left(x_{i}-\bar{x}\right)}^{3}}{X_{rms}^{3}}$

**Table 4 entropy-23-01704-t004:** Classifier accuracy with and without Hilbert spectrum entropy at different rotating speeds.

Classifier	Entropy On/Off	Accuracy (%) with Features at Different Operating Speeds (rpm)						
		500	600	700	800	900	1000	1100
Naïve Bayes	With entropy	**86.3**	**50**	**83.1**	**65.0**	**55.0**	**63.1**	**80.6**
	Without entropy	66.3	48.8	68.1	53.1	50.6	49.4	68.8
KNN	With entropy	**81.3**	**45.6**	**58.8**	**48.1**	**51.9**	**51.9**	**76.9**
	Without entropy	66.3	43.1	54.4	42.5	47.5	41.9	57.5
SVM	With entropy	**82.5**	**51.2**	**83.1**	**67.5**	**62.5**	**61.9**	**81.9**
	Without entropy	64.4	48.8	63.7	55.6	62.5	46.9	72.5

**Table 5 entropy-23-01704-t005:** Classifier accuracy with multi-speed entropy.

Classifier	Accuracy (%)
Naïve Bayes	90.6
KNN	89.4
SVM	92.5

## Data Availability

Data is contained within the article.

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
