# Peer review of "Experimental Investigation and Fault Diagnosis for Buckled Wet Clutch Based on Multi-Speed Hilbert Spectrum Entropy"

_entropy, 2021, doi:10.3390/e23121704_

Round 1

Reviewer 1 Report

-The paper should be interesting ;;;
-it is a good idea to add a block diagram of the proposed research;;;
-What is the result of the analysis?;;
-please add photos of the application of the proposed research, 2-3 photos ;;; 
-figures should have high quality;;;
-what will society have from the paper?;;
-please compare the proposed analysis with thermal analysis for example:
 "Ventilation Diagnosis of Angle Grinder Using Thermal Imaging",
-references should be from the web of science 2019-2021 (50% of all references)
-Conclusion: point out what have you done;;;;
-please add some sentences about future work;;;

Author Response

Dear reviewer,

My response are listed in the word file. thank you for your effort and suggestion. 

Reviewer 2 Report

The paper presents a new fault diagnosis method based on Hilbert spectrum entropy to classify the buckling state of the wet clutch. A brief introduction of rub-impact among the clutch disks is made. Then the experimental objects are classified into four groups based on the health states. A bench test is realized in order to collect the vibration signal at varying operation speed. The raw data is processed via Hilbert-Huang transform to generate the Hilbert spectrum and corresponding time-frequency entropy. A fault diagnosis method for the multidisc wet clutch system based on Naïve Bayes is used finally.

The paper presents the method in detail, it is based on experimental data taken from the test stand. But the results of the processing do not clearly show the existence or not of a defect, the differences between the normal case and the defects being almost imperceptible in most of the analyzed cases.

Much of the bibliography of the paper is recent. Also, of note figure 6, with the test stand, very clearly presented.

The text includes many phrases and ideas separated only by commas. They should be reformulated in short and clear sentences. Subchapter 3.1 is very briefly described, while 4.1 is given in more detail.

To be corrected:

Page 1, line 20 – “…feature inputs. the result demonstrates…”

Page 1, line 26 – “The In the transmission system…”

Page 2 – Figure 1  - line 53 - to improve the quality of the figure; line 54 – “Figure1” (space required)

Page 3, lines (95-106) – to be reformulated, because what is described in points (1) - (3) does not represent «some major contributions» of the work!

Page 3, lines 110-111 «op-eration”

Page 3, line 118 – “The This paper mainly adopts…”

Page 3, line 124 – Figure 3 – Figure caption should remain on the same page.

Page 4, line 143 – “5 c),”

Page 7, line 194 – “shown in 'figure 8' ”

Page 7 – The lines 193-194 and 195-196 repeat the same idea.

Page 13 – There are two figures 13.

Page 15 – lines 399-401 – to check and correct the reference [11] and its citation in the text (page 2).

Author Response

(The authors gave the same response as above.)

Reviewer 3 Report

This paper focus on a fault diagnosis approach to identify the buckling state for the wet clutch. It is based on Hilbert spectrum entropy to classify the buckling state of the wet clutch.

The paper deals with an interesting topic, although some minor and major observations must be addressed before a final decision can be taken.

1.- English grammar and style require a deep revision. It makes the paper difficult to follow.

2.- Line 25, please revise ”The In the transmission system”

3.- Line 118, please revise “The This paper mainly adopts ”

4.- Please improve the quality of Figure 3

5.- Used sensors and measuring devices. Full information is required

6.- It is not clear why the Hilbert Huang transform is applied. Wavelets and other time-frequency transforms are also appropriated to solve this problem.

7.- Different features are obtained for classification. I highly believe that a classification startegy such as LDA (or CVA) + kNN could help in increasing classification accuracy, please try.

The Reviewer suggests revising the work based on the suggestions above in order to improve its readability, scientific interest and quality.

Author Response

(The authors gave the same response as above.)

Round 2

Reviewer 1 Report

The paper is corrected according to my comments.

Author Response

Dear reviewer, 

Thank you for your time and advice.

Kind regards,

Jiaqi Xue

Reviewer 2 Report

It is noted that the authors have made efforts to improve the work, both technically and in terms of English. The explanations are clearer and the figures are improved. However, a few more corrections are needed:

Pag.5, lines 192-193 - incomplete sentence

Pag.12, line 279 - What do the figures represent? (there are probably two figures no. 11)

In Fig. 11b (page 12), please specify why the representation includes the 1kHz-3kHz frequency range. Was the entropy calculated only for this interval? Why?

Pag. 14 – lines 298-299 – to correct “The present work has calculated the time-frequency entropy”

Pages 15-16 – fig. 13. It is better to emphasize what each figure represents and the differences between them.

Author Response

Dear reviewer, 

My response to your comments are attached. 

Reviewer 3 Report

The author has replied all my questions

Author Response

Dear reviewer, 

Thank you for your time and advice.

Kind regards,

Jiaqi Xue

This manuscript is a resubmission of an earlier submission. The following is a list of the peer review reports and author responses from that submission.